# Effects of Different Blanching Methods on the Quality of *Tremella fuciformis* and Its Moisture Migration Characteristics

**DOI:** 10.3390/foods12081669

**Published:** 2023-04-17

**Authors:** Zhipeng Zheng, Li Wu, Yibin Li, Wei Deng, Shouhui Chen, Hongbo Song

**Affiliations:** 1College of Food Science, Fujian Agriculture and Forestry University, Fuzhou 350002, China; 2Institute of Agricultural Engineering Technology, Fujian Academy of Agricultural Sciences, Fuzhou 350003, China; 3Key Laboratory of Subtropical Characteristic Fruits, Vegetables and Edible Fungi Processing (Co-Construction by Ministry and Province), Ministry of Agriculture and Rural Affairs, Fuzhou 350003, China; 4Fujian Province Key Laboratory of Agricultural Products (Food) Processing Technology, Fuzhou 350003, China

**Keywords:** *Tremella fuciformis*, ultrasonic-low temperature blanching, quality, moisture migration characteristics

## Abstract

Blanching is a critical step in the processing of *Tremella fuciformis* (*T. fuciformis*). The effects of different blanching methods (boiling water blanching (BWB), ultrasonic-low temperature blanching (ULTB), and high-temperature steam (HTS)) on the quality and moisture migration characteristics of *T. fuciformis* were investigated. The results showed that the *T. fuciformis* blanched by ULTB (70 °C, 2 min, 40 kHz, 300 W) had the best quality, including a brighter appearance, superior texture, and good sensory features, with a polysaccharide content of 3.90 ± 0.02%. The moisture migration characteristics of *T. fuciformis* after blanching exhibited four peaks, displayed strong and weak chemically bound water, immobilized water, and free water, whereas ULTB had a weak effect on the freedom of water in *T. fuciformis*. The study will provide the foundation for the factory processing of *T. fuciformis*.

## 1. Introduction

*Tremella fuciformis* (*T. fuciformis*) is a well-known edible fungus widely produced in China, which belongs to the order of Tremellales and the family of *Tremellaceae* [1,2]. *T. fuciformis* is rich in physiologically active substances and is often defined as a tonic and medicine. Thus, it has attracted the attention of consumers and researchers [3,4]. The nutritional value and pharmacological activities of *T. fuciformis* have been intensively studied, but its value in food processing has been little studied [5]. *T. fuciformis* can improve the sensory attributes and physicochemical properties of foods, often as a meat substitute or fat substitute and also as a supplement to bread [6]. Hu [7] et al. used *T. fuciformis* to replace 75% of the pork fat in pork sausages and found that the pork sausages had the best organoleptic properties. Lin [8] et al. added *T. fuciformis* polysaccharides (TFPS) to low-fat yogurt, which increased the solids content and water-holding capacity of the yogurt, and had positive effects on sensory and texture.

Blanching is a crucial step in the processing of *T. fuciformis*. Few studies have been conducted to investigate the quality of the blanched *T. fuciformis* fruiting body, especially on the treatment of the fungus with different processing methods. Fresh *T. fuciformis* undergoes strong metabolic activities and respiration under the action of various enzymes and is susceptible to microbial contamination, which affects food value and commercial appearance [9]. Thermal processes of fresh *T. fuciformis* could reduce the abundance of microorganisms and inhibit the activity of enzymes from increasing their safety [10]. Boiling water blanching (BWB) and high-temperature steam (HTS) are the most common methods of thermal processes, which often lead to quality loss and nutrient loss [11,12]. Steam is a versatile heating medium that can retain the original color of food and has little effect on the quality of food [13,14]. Ju [15] et al. processed *Inonotus Obliquus* with steam and found that soluble phenolic content and antioxidant activity were enhanced. Thermal processes at a lower temperature (70 °C) can maintain food quality and inhibit the activity of most enzymes [16]. In addition, ultrasound, an auxiliary processing method, could improve the texture, sensory quality, and content of organic taste compounds and inactivate microorganisms [17,18,19]. Ganjdoost [20] et al. processed *Agaricus Bisporus* in an ultrasonic water bath and found that the quality was improved, and the extended shelf life was extended. Mushroom slices were treated with ultrasound (28 kHz, 600 W), which reduced the drying time by 21.43% and obtained a superior texture [21]. Ultrasound combined thermal processes at the temperature of 70 °C was a potential method and rarely used in the processing of *T. fuciformis*. In addition, the high moisture content may lead to decreased quality and microbial contamination and a significant reduction in shelf life [22,23,24]. The moisture migration during the *T. fuciformis* process has not been reported.

Therefore, this paper aims to investigate the effects of different blanching methods, such as boiling water blanching (BWB), ultrasonic-low temperature blanching (ULTB), and high-temperature steam (HTS), on the quality of *T. fuciformis*, including color, texture, polysaccharide content, and sensory features, and to study their effects on the water migration characteristics. It was thus expected to provide a basis for the industrial processing of *T. fuciformis*.

## 2. Materials and Methods

### 2.1. Materials

*T. fuciformis* was supplied by Fujian Xiangyun Biotech Development Co., Ltd. (Fuzhou, Fujian, China). The *T. fuciformis* specimens, featuring an intact body, uniform size, and a smooth surface, were utilized in the experiments. All other chemicals used were of analytical grade.

### 2.2. Blanching Methods of T. fuciformis

The stem was removed from *T. fuciformis* and, the fruiting body was cut into small pieces with a width of 3 cm. BWB treatment was that pieces were blanched in boiling water at a solid-to-liquid ratio of 1:10 (w:v). ULTB treatment was performed in an ultrasonic cleaner (40 kHz, 300 W, 70 °C) at a solid-to-liquid ratio of 1:10 (w:v). HTS treatment was that the pieces were laid flat in a tray blanched by water vapor. The unblanched *T. fuciformis* piece was defined as the CK group or the 0 min group.

### 2.3. Chromatism

The surface chromatism of the fruiting body was determined by an NS810 colorimeter (Threenh Technology Co., Ltd. Shenzhen, China). The color was expressed in terms of 3 parameters, *L** (lightness), *a** (redness), and *b** (yellowness) values.

### 2.4. Texture Profile Analysis

The textural properties of blanched samples were analyzed using a TA. XT Texture Analyzer (Stable Micro Systems, Surrey, UK) equipped with a P/36R probe; 30 g homogenized sample was played into a 100 mL beaker, trigger force = Auto, pre-test speed = 1 mm/s, test speed = 2 mm/s, post-test speed = 2 mm/s, and the compression = 50%.

### 2.5. Polysaccharide Content

The polysaccharide content in the blanched samples was determined using the phenol-sulphuric acid method. The sample was dried in an air dryer to a constant weight and then immersed in hot water (100 °C) at a solid-to-liquid ratio of 1:150 (w:v) for 2 h. Finally, the liquor was obtained after being filtered and further separated via centrifugation. The clarified liquor was volumized to 1000 mL, and 1 mL of the sample was measured using a TU-1810 UV-Vis spectrophotometer (Purkinje GENERAL Instrument Co., Ltd. Beijing, China) at 490 nm. Equation (1) was the glucose standard curve. The polysaccharide content was calculated by Equation (2).
(1)$$y=9.812x-0.016\;(R^{2}=0.9941)$$
(2)$$\omega =\frac{x\times 1000\times 0.9}{m}\times 10^{-4}\times 100\%$$
where *y* denotes the absorbance per mL of sample solution, and the *x* is the polysaccharide content per mL of sample solution (μg/mL), *ω* is the polysaccharide content of the sample (%), *m* is the weight of the sample, 0.9 denotes correction factor for glucose.

### 2.6. Moisture Migration Characteristics

Spin-spin T_2_ relaxation was determined using an NMI20-040H-I NMR analyzer (Suzhou Niumag Analytical Instrument Corporation, Suzhou, China) with resonance frequency for protons of 20.628 MHz at a constant temperature (32 °C). Five to six leaves of *T. fuciformis* were stacked in a diameter of 40 mm tube. The CPMG sequence parameters for 90° pulse and 180° pulse were 6.60 μs and 11.60 μs, respectively. The interval between two scans was 4000 ms, 3 repeat scans per sample, the echo time was 0.6 ms with 9000 echos, and CPMG data were fitted using the SIRT 1,000,000 algorithm.

After LF-NMR analysis, samples were scanned to obtain the pseudo-color T_2_-weighted images of *T. fuciformis* by NMI20-040H-I NMR analyzer with SE imaging sequence. Repetition time (TR) and echo time (TE) was set to 19.94 ms and 1300 ms, respectively; the slice thickness was 4.65 mm, and images were acquired as 3 scan repetitions. In a proton-weighted image, higher proton density means a stronger signal, so it presents brighter region in the image.

### 2.7. Sensory Evaluation

Sensory evaluation was performed by 10 panelists (five male and five female). They have been professionally trained before the sensory evaluation. All samples were packed in stainless steel trays and randomly numbered. Each panelist needed to remove residual taste with pure water before tasting subsequent samples. The sensory test included color, texture, aroma, appearance, and flavor, and the maximum score was 20 for each sensory attribute, with which score of 20 indicating very desirable and a score of 1 meaning very undesirable (The scoring criteria of the sensory evaluation was shown in the Appendix A).

### 2.8. Statistical Analysis

The data were reported as the mean ± standard deviation (SD). The measurements were subjected to ANOVA and Duncan’s test by SPSS 26.0 (IBM, Chicago, IL, USA); A value of *p* < 0.05 was regarded as statistically significant. Graphs were performed using Origin pro 2018 (Origin Lab Corporation, Northampton, MA, USA). Three groups of samples were measured, and the average was taken.

## 3. Results and Discussion

### 3.1. Chromatism

Chromatism is a pivotal parameter for evaluating the quality of *T. fuciformis*, as it reveals the efficacy of blanching techniques in preserving *T.* fuciformis’s color [25]. The changes of *L**, *a**, and *b** values for the BWB, ULTB, and HTS groups are shown in Table 1. With the increase in the blanching time in the BWB and HTS groups, the *L** decreased rapidly, while the *L** of the ULTB group increased in the first 2 min, followed by a decline; *L** of the ULTB group was significantly higher than that of the BWB and HTS groups at 1.5 min~2 min (*p* < 0.05). The *a** values of the ULTB and HTS groups showed a downward trend with increasing time, causing the surface color to gradually turn green. Conversely, the *a** of the BWB group was increased with time, which was different from the results of the ULTB and HTS groups. In addition, as the blanching time increased, the *b** values of the BWB and ULTB groups initially decreased and then increased. The ULTB group was significantly bluer than the other groups at 2 min~3 min (*p* < 0.05), which indicated that these two blanching methods induced a certain degree of blueness in *T. fuciformis* and then gradually decreased, probably because the surface color of *T. fuciformis* was destroyed by high temperature, and become transparent.

A significant difference (*p* < 0.05) was observed by the presence at the different time of the same method. It was worth noting that the color of the samples did not significantly different from the CK group (*p* > 0.05), indicating that the blanching methods maintained the color of *T. fuciformis*. The ULTB group (t = 2 min) was not significantly different from the CK group in *L** and *a**. Compared to the CK group, the surface color turned blue after blanching was an unavoidable trend. The results demonstrated that the ULTB treatment (t = 2 min) safeguarded the color of *T. fuciformis*.

### 3.2. Texture Profile Analysis (TPA)

The texture is a crucial aspect of the sensory experience of *T. fuciformis*, and over-processing could result in the heavy loss of its flavor [7,26]. The effects of different blanching methods on the textural properties of *T. fuciformis* were shown in Figure 1 and Figure 2. The textural properties of the BWB group decreased rapidly with increasing time. Probably because of the high temperature and the boiling water that destroyed the structure of *T. fuciformis* [27]. The textural properties of the HTS group displayed a significant decrease in the blanching time. The ULTB group (t = 1.5 min~2 min) was significantly higher than the other two groups in all textural properties (*p* < 0.05). The blanching method had an adverse effect on the texture of the *T. fuciformis*, but an upward trend was observed in the ULTB group (t = 1.5 min), which was not significantly different from the CK group in terms of firmness and consistency (*p* > 0.05), this result indicated that ultrasound had the potential to enhance the textural properties, but the textural properties of the ULTB group (t = 2.5 min~3 min) decreased rapidly, probably because the prolonged ultrasound destroyed the *T. fuciformis* [28].

According to the data from the TPA experiments, we found that the ULTB group (t = 1 min~2 min) exhibited superior texture properties for a short time. Firmness is a main factor related to brittleness, and blanching methods significantly affected the firmness of the *T. fuciformis*; the ULTB group demonstrated higher firmness, but higher firmness does not necessarily equate to better taste [29]. Furthermore, viscosity was negatively affected during the tasting process regarding *T. Fusiformis*, while the ULTB group possessed a higher viscosity. This might be due to the homogenization of *T. Fusiformis* before the TPA experiment, and the ULTB treatment had a minor effect on its textural properties. The sample absorbed less water than the other two groups, and the homogenized liquid was thicker, resulting in higher viscosity [30,31].

### 3.3. Polysaccharide Content

Polysaccharide, the most valuable active constituents of *T. fuciformis*, have pharmacological activities, including anti-tumor, anti-diabetic, and hypolipidemic [32,33]. The polysaccharide content of *T. fuciformis* with different blanching methods was shown in Figure 3. The results indicated that the BWB group had the best polysaccharide extraction rate, which was significantly higher than the ULTB and HTS groups, except at 2 min. (*p* < 0.05), probably because boiling water was helpful in the extraction of polysaccharide. This finding was consistent with previous results by Chen [1] for the optimization of the extraction process of polysaccharide from *T. fuciformis*. The ULTB group showed an increasing trend in polysaccharide content before 2 min, indicating that ultrasound could improve the extraction rate of polysaccharides from *T. fuciformis*. This result is in agreement with previous experiments [34]. However, the polysaccharide content of the ULTB group decreased rapidly after 2 min, possibly due to prolonged ultrasound, which resulted in ultrasonic cavitation and reactive free radicals. The macromolecular chains of polysaccharides in *T. fuciformis* were disrupted and degraded, resulting in the loss of polysaccharide content during processing [35]. In contrast, the polysaccharide content of the HTS group decreased rapidly, possibly due to excessive steaming, leading to rapid softening of *T. fuciformis* and loss of active constituents [36,37].

As shown in Figure 3b, we found no difference in the BWB group (t ≤ 1.5 min), ULTB group (t = 2 min), and CK group (*p* > 0.05). However, we found that the BWB treatment was not the appropriate method in the TPA analysis. The ULTB group (t = 2 min) was significantly higher than the different time in the same blanching method (*p* < 0.05). Therefore, the ULTB group (t = 2 min) was a more effective method that could retain the polysaccharide content of *T. fuciformis*.

### 3.4. Moisture Migration Characteristics

The study examined the effects of different blanching methods on the moisture migration characteristics of *T. fuciformis*. The transverse relaxation time of *T. fuciformis* through different blanching methods was investigated, and the results were presented in Figure 4. Based on the transverse relaxation time, four different molecular environments of water components in *T. fuciformis* were identified and respectively marked as T_2b_, T_21_, T_22_, and T_23_, starting from the left. T_2b_ and T_21_ are strong chemically bound water and weak chemically bound water associated with cell wall polysaccharide, respectively [38]; T_22_ represents immobilized water that interacts with some macromolecular proteins in the cytoplasm, T_23_ represents free water with high mobility in the vacuole, and a small peak following T_23_, which is part of the water precipitated from the processing of *T. fuciformis*, the shorter the relaxation time T_2_ was, the more tightly the water binds to the substrate, the longer the relaxation time T_2_ was, the greater the degree of moisture fluidity was [39,40,41,42].

From the results, only three peaks, T_2b_, T_22,_ and T_23_, were observed in the CK group, and T_22_ and T_23_ were tightly bound. After treatment with three blanching methods, T_21_ was detected and increased water mobility to varying degrees. Free water was the main water component of *T. fuciformis*, and all three methods increased its amplitude intensity with blanching time. The T_23_ peak of the BWB group shifted significantly to the right side with increasing blanching time, and there was a tendency for T_22_ to merge with T_23_; these findings indicated that the BWB group increased the freedom of water [43], the high moisture freedom made *T. fuciformis* susceptible to spoilage after processing [44]. Additionally, the water precipitated in the BWB group was very obvious, probably because the high temperature accelerated the destruction of the texture of the *T. fuciformis*. The relaxation time change of the HTS group was smaller than that of the BWB group, but the amplitude increased significantly with blanching time. The relaxation time of the T_22_ and T_23_ shifts in the ULTB group was insignificant and shorter than in the BWB and HTS groups. This indicates that ULTB treatment had less effect on the water freedom and was an effective method for maintaining the quality of *T. fuciformis*.

The pseudo-color T_2_-weighted images of *T. fuciformis* under different blanching methods were presented in Figure 5, which visually represented the changes in moisture content. In these images, the red areas correspond to high moisture content [45]. It could be noticed that the red area of the cross-section of the BWB group increased significantly, indicating an increase in the proportion of free water in *T. fuciformis* and an enhancement in the freedom of water. The decrease in brightness of the BWB group could be attributed to the high water content that made the leaves appear transparent and dim, as indicated by the chromaticity data (Table 1). Furthermore, the red area of the HTS group also increased with time. In contrast, the red area of the ULTB group increased to a lesser extent, implying that ULTB maintained a greater degree of water composition in *T. fuciformis*, which was consistent with the conclusion obtained by LF-NMR. Notably, due to the high water content, the original shape of leaves was lost in the BWB group, which indicated that the textural properties were destroyed, and they tended to absorb water continuously.

### 3.5. Sensory Evaluation

The results of the sensory evaluation of *T. fuciformis* subjected to various blanching methods in Table 2. The BWB group exhibited a substantial decrease in sensory score with processing time, with a flavor score of only 13.00 ± 0.47 at 3 min; the result indicated that boiling water blanching was an adverse effect on the sensory attributes of the *T. fuciformis*, the texture properties of the sample were destroyed, and sensory attributes become unacceptable [46,47]. The HTS group had a high sensory score at 1 min, then decreased rapidly in its texture, appearance, and flavor scores. The score of its appearance at 1 min was 17.3 ± 0.48, which was significantly higher than the BWB and ULTB groups (*p* < 0.05), probably because it was not blanched in water and the brief steam treatment preserved its texture and sensory attributes [48,49]. The ULTB group showed better sensory scores than the other groups, with the highest sensory score at 2 min, which was significantly higher than the other two groups (*p* < 0.05), indicating that the ULTB treatment was superior in terms of sensory properties. In terms of appearance and aroma, the scores of the ULTB group (t = 2 min) were significantly higher than those of the same treatment methods at different time, and also had the highest score in terms of color, texture, and flavor. However, it differed from the best texture results obtained by the ULTB group at 1.5 min in texture profile analysis, and this was likely due to the insufficient low-temperature blanching time and higher hardness, which affected the sensory scores. Sensory evaluation is a technique used to determine the acceptability of a food product based on consumer perceptions [50]. The results demonstrated that had a positive effect on the sensory attributes of *T. fuciformis* and *T. fuciformis* treated with ULTB was preferred by the tasters.

## 4. Conclusions

In this study, the effects of boiling water blanching (BWB), ultrasonic-low temperature blanching (ULTB), and high-temperature steam (HTS) on the processing quality and moisture migration characteristics of *T. fuciformis* were analyzed. Compared to BWB and HTS treatments, the results showed that *T. fuciformis* treated with ULTB (t = 2 min) maintained suitable chromatism, texture, and higher polysaccharide content than the HTS and BWB groups and obtained the highest sensory score with a score of 87.0 ± 1.83. Based on the transverse relaxation time, four different molecular environments of water components in *T. fuciformis* were identified, named strong and weak chemically bound water, immobilized water, and free water, respectively. The three blanching methods all increased the freedom of water to varying degrees with processing time, while ULTB had a weak effect on the freedom of water of *T. fuciformis*, which was in accordance with the pseudo-color T_2_-weighted images. Based on these results, we determined that the optimal blanching method was ultrasonic-low temperature blanching (ULTB, 70 °C, 2 min, 40 kHz, 300 W). Through this blanching method, the overall nutritional quality and sensory quality of *T. fuciformis* after processing could be improved, and the degree of the freedom of the four different molecular environments of water components in *T. fuciformis* has little change.

## Figures and Tables

**Figure 1 foods-12-01669-f001:**
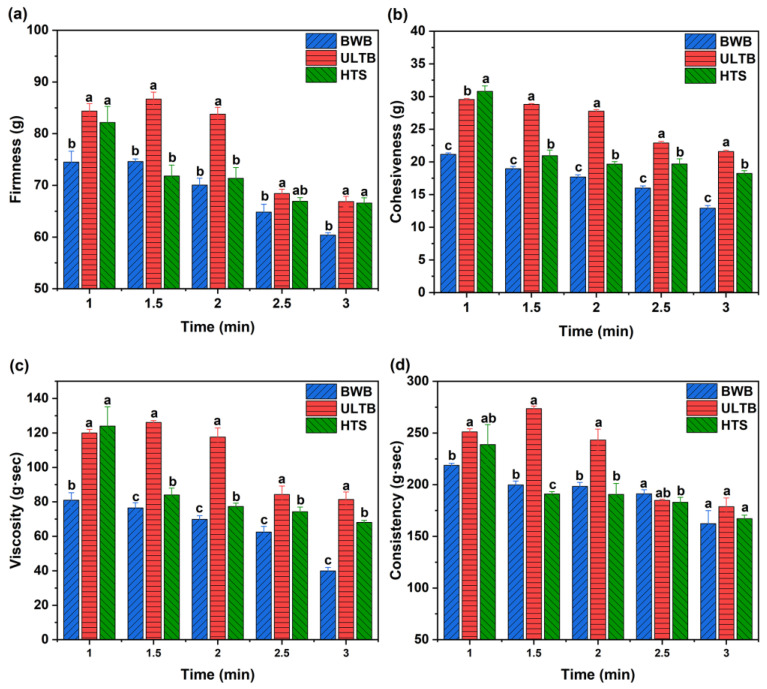
Effect of different blanching methods on the textural properties of *T. fuciformis*. Note: (**a**) Firmness, (**b**) Cohesiveness, (**c**) Viscosity, (**d**) Consistency. The different lowercase letters indicate significant differences among the three treatments at the same time (*p* < 0.05).

**Figure 2 foods-12-01669-f002:**
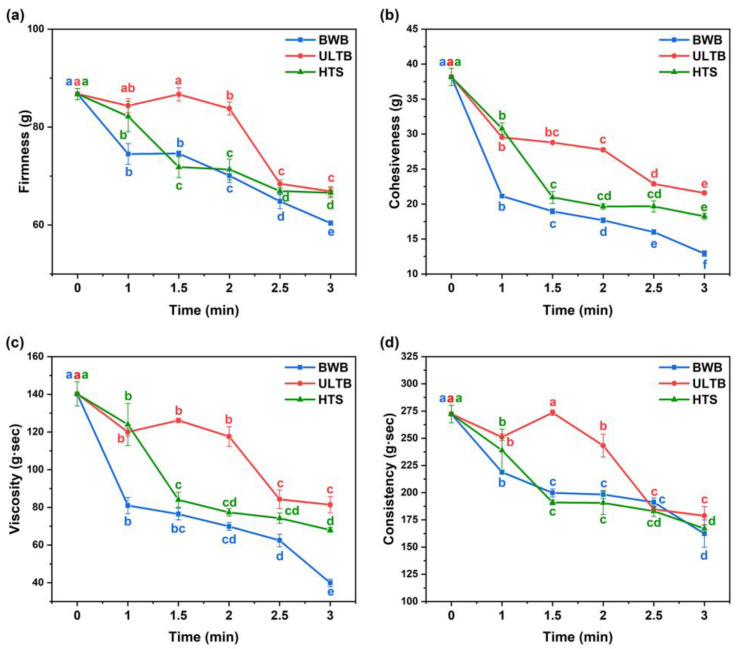
Dynamics of the texture of *T. Fusiformis* treated with different blanching methods. Note: (**a**) Firmness, (**b**) Cohesiveness, (**c**) Viscosity, (**d**) Consistency. The different lowercase letters with the same color indicate significant differences between the different time at the same treatment (*p* < 0.05).

**Figure 3 foods-12-01669-f003:**
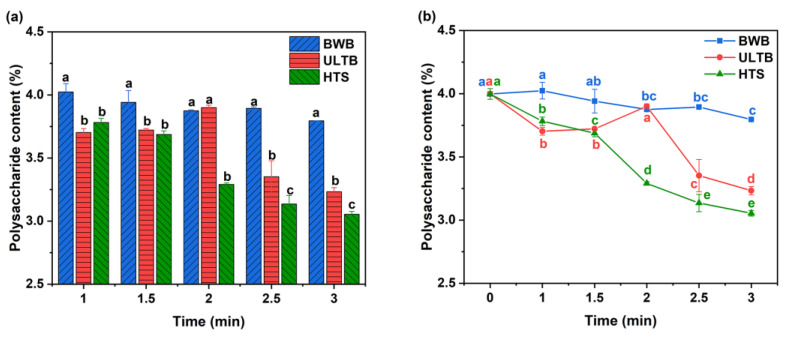
Polysaccharide content of different blanching methods of *T. fuciformis*. Note: (**a**) The different lowercase letters indicate significant differences among the three treatments at the same time (*p* < 0.05). (**b**) The different lowercase letters with the same color indicate significant differences at different time at the same treatment (*p* < 0.05).

**Figure 4 foods-12-01669-f004:**
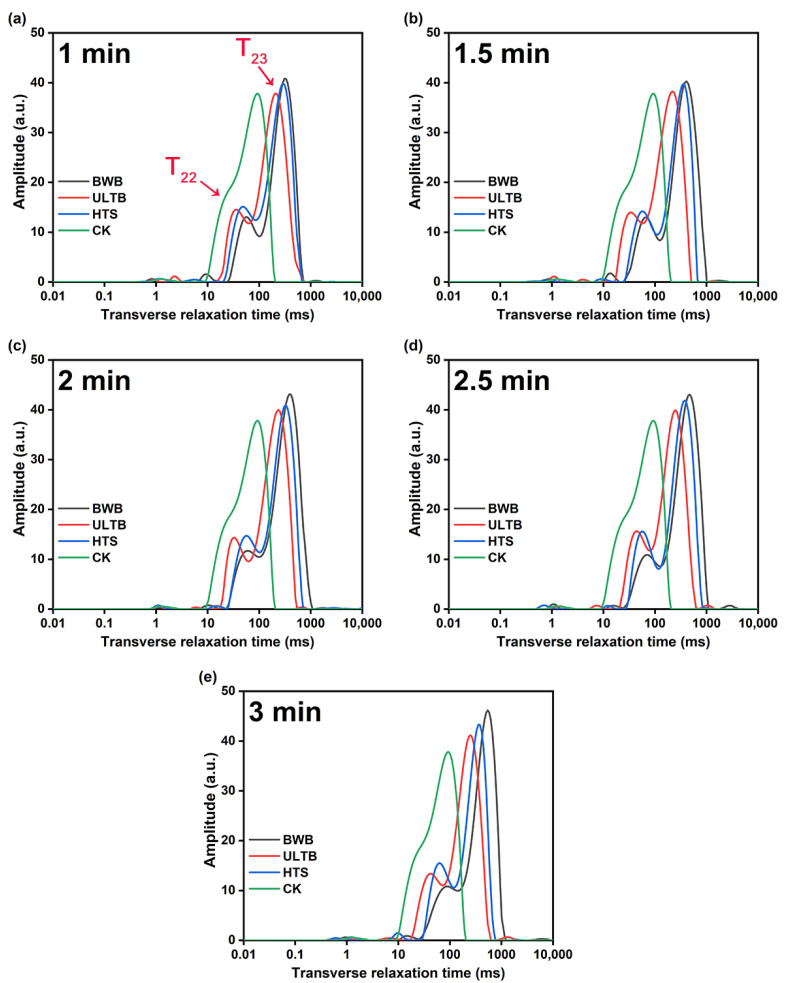
Change of transverse relaxation time and relative amplitude of *T. fuciformis* with different blanching methods: (**a**) 1 min, (**b**) 1.5 min, (**c**) 2 min, (**d**) 2.5 min, (**e**) 3 min.

**Figure 5 foods-12-01669-f005:**
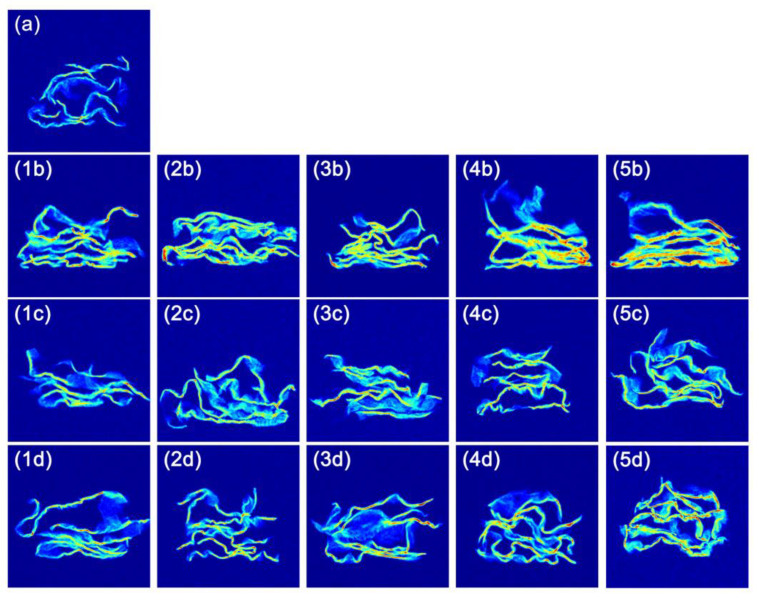
The pseudo-color T_2_-weighted images of *T. fuciformis* with different blanching methods. (**a**) The pseudo-color T_2_-weighted image of fresh *T. fuciformis*. (**1b**–**5b**) The pseudo-color T_2_-weighted images of BWB group at 1 min~3 min. (**1c**–**5c**) The pseudo-color T_2_-weighted images of ULTB group at 1 min~3 min. (**1d**–**5d**) The pseudo-color T_2_-weighted images of HTS group at 1 min~3 min.

**Table 1 foods-12-01669-t001:** Chromatism of different blanching methods of *T. fuciformis*.

Chromatism	Blanching Time	Blanching Methods								
		BWB			ULTB			HTS		
			*	#		*	#		*	#
*L**	0 min	42.21 ± 0.85		a	42.21 ± 0.85		ab	42.21 ± 0.85		a
	1 min	40.66 ± 1.22	a	ab	40.57 ± 1.01	a	c	41.90 ± 0.59	a	ab
	1.5 min	40.45 ± 0.52	b	b	41.53 ± 0.28	a	abc	40.91 ± 0.38	ab	bc
	2 min	39.90 ± 0.72	b	b	42.56 ± 0.29	a	a	40.36 ± 0.50	b	c
	2.5 min	39.84 ± 0.90	b	b	41.11 ± 0.37	a	bc	40.23 ± 0.33	ab	c
	3 min	39.61 ± 0.37	b	b	40.62 ± 0.25	a	c	40.18 ± 0.30	ab	c
*a**	0 min	−0.98 ± 0.41		a	−0.98 ± 0.41		b	−0.98 ± 0.41		c
	1 min	−1.00 ± 0.07	b	a	−0.82 ± 0.02	a	a	−0.86 ± 0.03	a	a
	1.5 min	−0.97 ± 0.04	b	a	−0.84 ± 0.04	a	a	−0.88 ± 0.02	a	ab
	2 min	−0.95 ± 0.05	a	a	−0.99 ± 0.03	a	b	−0.94 ± 0.06	a	abc
	2.5 min	−0.95 ± 0.04	a	a	−1.00 ± 0.04	a	b	−0.97 ± 0.05	a	bc
	3 min	−0.94 ± 0.01	a	a	−1.02 ± 0.01	b	b	−0.98 ± 0.06	ab	c
*b**	0 min	−0.39 ± 0.26		a	−0.39 ± 0.26		a	−0.39 ± 0.26		a
	1 min	−1.10 ± 0.08	b	bc	−0.62 ± 0.09	a	b	−0.92 ± 0.08	b	b
	1.5 min	−1.35 ± 0.11	b	d	−1.25 ± 0.02	b	c	−1.03 ± 0.09	a	b
	2 min	−1.28 ± 0.14	b	cd	−1.50 ± 0.08	c	d	−1.04 ± 0.09	a	b
	2.5 min	−1.23 ± 0.10	b	bcd	−1.59 ± 0.10	c	d	−1.02 ± 0.07	a	b
	3 min	−1.08 ± 0.08	a	b	−1.50 ± 0.04	b	d	−1.06 ± 0.09	a	b

Note: “*” significant difference among blanching methods at the same time, “#” significant difference among treatment time at the same blanching method. Different letters indicate significant differences (*p* < 0.05).

**Table 2 foods-12-01669-t002:** The sensory scores of *T. fuciformis* in different blanching methods.

Parameter	Blanching Time	Blanching Methods								
		BWB			ULTB			HTS		
			*	#		*	#		*	#
Color	1 min	15.80 ± 0.42	a	a	15.40 ± 0.70	ab	bc	14.90 ± 0.57	b	a
	1.5 min	15.60 ± 0.70	a	a	16.10 ± 0.74	a	ab	14.70 ± 0.82	b	a
	2 min	15.50 ± 0.71	b	a	16.70 ± 0.82	a	a	15.00 ± 0.47	b	a
	2.5 min	15.30 ± 0.48	b	a	16.20 ± 0.63	a	a	15.20 ± 0.79	b	a
	3 min	15.30 ± 0.67	a	a	15.20 ± 0.42	a	c	14.80 ± 0.79	a	a
Texture	1 min	15.60 ± 0.84	b	a	16.60 ± 0.84	a	c	16.60 ± 0.97	a	a
	1.5 min	16.10 ± 0.74	b	a	17.50 ± 0.71	a	ab	15.50 ± 1.18	b	b
	2 min	14.20 ± 0.79	c	b	17.80 ± 0.79	a	a	15.40 ± 0.70	b	b
	2.5 min	13.70 ± 0.67	b	b	16.70 ± 0.67	a	bc	14.20 ± 0.79	b	c
	3 min	13.60 ± 0.84	b	b	15.90 ± 0.74	a	c	13.20 ± 0.63	b	d
Aroma	1 min	16.20 ± 0.63	a	a	16.00 ± 0.82	a	b	16.70 ± 0.48	a	a
	1.5 min	16.00 ± 0.47	b	a	16.90 ± 1.00	a	b	17.20 ± 0.63	a	a
	2 min	15.00 ± 0.94	c	b	17.90 ± 0.74	a	a	17.00 ± 0.67	b	a
	2.5 min	15.00 ± 0.67	b	b	16.90 ± 0.88	a	b	16.90 ± 0.57	a	a
	3 min	14.30 ± 0.67	b	b	16.50 ± 0.53	a	b	16.80 ± 0.42	a	a
Appearance	1 min	16.10 ± 0.99	b	a	16.20 ± 1.03	b	b	17.30 ± 0.48	a	a
	1.5 min	16.00 ± 1.05	a	a	16.30 ± 0.67	a	b	16.20 ± 0.79	a	a
	2 min	14.10 ± 0.74	c	b	17.50 ± 0.97	a	a	16.20 ± 0.92	b	a
	2.5 min	14.30 ± 0.67	b	b	16.10 ± 0.57	a	b	15.90 ± 0.88	a	a
	3 min	13.50 ± 0.71	b	b	16.00 ± 0.67	a	b	15.90 ± 0.74	a	a
Flavor	1 min	16.10 ± 0.88	a	a	16.50 ± 0.97	a	ab	16.70 ± 0.67	a	a
	1.5 min	15.80 ± 0.79	a	a	16.30 ± 0.83	a	bc	16.50 ± 0.71	a	a
	2 min	13.80 ± 1.03	c	b	17.10 ± 0.57	a	a	15.40 ± 0.84	b	b
	2.5 min	13.40 ± 0.84	b	b	16.00 ± 0.47	a	bc	14.10 ± 0.74	b	c
	3 min	13.00 ± 0.47	b	b	15.70 ± 0.48	a	c	13.40 ± 0.84	b	c

Note: “*” significant difference among blanching methods at the same time, “#” significant difference among treatment time at the same blanching method. Different letters indicate significant differences (*p* < 0.05).

## Data Availability

Data is contained within the article or Appendix A.

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
