# Peer review of "Effects of Different Blanching Methods on the Quality of Tremella fuciformis and Its Moisture Migration Characteristics"

_foods, 2023, doi:10.3390/foods12081669_

Round 1

Reviewer 1 Report

I have reviewed ‘Effects of different blanching methods on the quality of Tremella Fuciformis and its moisture migration characteristics’. Overall, the paper is very interesting, and the experiment is well designed with sufficient data to support. It could provide valuable information to the food science area. However, the language is poor and have several grammar issues. Some minor comments:

1.     Page 2 line 47: blanching is an important unit operation for …

2.     Section 2.2: grammar issue with some sentences, please revise accordingly.

3.     Page 3 line 123: please change reviewers to panelists.

4.     For the numbers in all tables, no need to keep so many significant figures.

Author Response

Reviewer 1

I have reviewed ‘Effects of different blanching methods on the quality of Tremella Fuciformis and its moisture migration characteristics’. Overall, the paper is very interesting, and the experiment is well designed with sufficient data to support. It could provide valuable information to the food science area. However, the language is poor and have several grammar issues. Some minor comments:

Our manuscript entitled, “Effects of different blanching methods on the quality of Tremella Fuciformis (T. Fuciformis) and its moisture migration characteristics” has been carefully revised according to reviewer’s suggestions. About the English writing of the manuscript, we ask for native English speaker to revise the paper before it was submitted to the magazine. Now I answer the questions one-by-one.

Q1. Page 2 line 47: blanching is an important unit operation for …

Response: Thanks for the reviewer’s comment. We are very sorry for our incorrect grammar (Page 2 Line 47), “Blanching is an important unit for T. Fuciformis processing” had been corrected as “Blanching is a crucial step in the processing of T. Fuciformis.”

Q2. grammar issue with some sentences, please revise accordingly.

Response: Thanks for the reviewer’s comment. We apologize for the grammar issue in the Section 2.2 of the original manuscript and revised as follows: “The stem was removed from T. Fuciformis and the fruiting body was cut into small pieces with a width of 3 cm. BWB treatment was that pieces were blanched in boiling water at a solid-to-liquid ratio of 1:10 (w:v). ULTB treatment was performed in an ultrasonic cleaner (40 kHz, 300 W, 70℃) at a solid-to-liquid ratio of 1:10 (w:v). HTS treatment was that the pieces were laid flat in a tray blanched by water vapor. The unblanched T. Fuciformis piece was defined as the CK group or the 0 min group. ”

Q3.Page 3 line 123: please change reviewers to panelists.

Response: Thanks for the reviewer’s comment. We are very sorry for our incorrect writing. We have changed reviewers to panelists in the section 2.7.

Q4. For the numbers in all tables, no need to keep so many significant figures.

Response: Thank you for your suggestion. We have modified these tables in accordance with the reviewer’s comment, the numbers in all tables have been kept two decimal figures. We appreciate for reviews’ warm work earnestly and hope that the correction will meet with approval.

Reviewer 2 Report

The research is interesting and worthwhile. There are a few aspects that I would like greater clarification on. The choice of processes is important and perhaps you can explain how you reached this choice, for instance discussing a little more about thermal and non thermal processes which are used for the extension of mushroom shelf life and their effect on quality and composition (Zheng, Q., Gao, P., Liu, T., Gao, X., Li, W. and Zhao, G. (2022), Effects of drying methods on colour, amino acids, phenolic profile, microstructure and volatile aroma components of Boletus aereus slices. Int J Food Sci Technol, 57: 5164-5174; A. Rawson, B. K. Tiwari, N. Brunton, C. Brennan, P. J. Cullen & C. P. O'Donnell (2012) Application of Supercritical Carbon Dioxide to Fruit and Vegetables: Extraction, Processing, and Preservation, Food Reviews International, 28:3, 253-276, DOI: 10.1080/87559129.2011.635389)

The effect of temperature on enzyme activity is also well known and perhaps you can discuss this in relation to thermal and non-thermal treatments and how these affect colour, texture and sensory properties (Umair, M., Jabbar, S., Lin, Y., Nasiru, M.M., Zhang, J., Abid, M., Murtaza, M.A. and Zhao, L. (2022), Comparative study: Thermal and non-thermal treatment on enzyme deactivation and selected quality attributes of fresh carrot juice. Int J Food Sci Technol, 57: 827-841; Hasheminya, S.-M. and Dehghannya, J. (2022), Non-thermal processing of black carrot juice using ultrasound: Intensification of bioactive compounds and microbiological quality. Int J Food Sci Technol, 57: 5848-5858.)

One question I do have about the methodology is why only 10 people were used for the sensory analysis ? The sample size seems to be very small. Could you elaborate on the training that these individuals were given please ?

Also, can you please look at the graphs and tables, and pay attention to the statistical analysis conducted- re-run the analysis. There appears to be some results where the values of the readings appear to be significantly different, but the analysis does not show this up. Please pay careful attention to this and when describing the results please concentrate on only those which are significantly different

Author Response

Reviewer 2

The research is interesting and worthwhile. There are a few aspects that I would like greater clarification on.

Q1. The choice of processes is important and perhaps you can explain how you reached this choice, for instance discussing a little more about thermal and non thermal processes which are used for the extension of mushroom shelf life and their effect on quality and composition.

Response: Thank you very much for your careful review and constructive suggestions with regard to our manuscript. It is really true as the reviewer pointed out that we should explain the reasons for the choice of processes. Thermal processes of fresh T. Fuciformis could reduce the abundance of microorganisms and inhibit the activity of enzyme to increase its safety. Boiling water blanching (BWB) and high temperature steam (HTS) are the most common methods of thermal processes. In addition, ultrasound could improve the texture and sensory quality. BWB, HTS and ultrasound are easy to be applied in industrial production for their low cost, so we choose the three processing methods. “Fresh T. fuciformis undergoes strong metabolic activities and respiration under the action of various enzymes and is susceptible to microbial contamination, which affects food value and commercial appearance [9]. Thermal processes of Fresh T. Fuciformis could reduce the abundance of microorganisms and inhibit the activity of enzyme to increase its safety [10]. Boiling water blanching (BWB) and high temperature steam (HTS) are the most common methods of thermal processes, which often lead to quality loss and nutrient loss [11,12]. Steam is a versatile heating medium, which can retain the original color of food and has little effect on the quality of food [13,14]. Ju [15] et al. processed Inonotus Obliquus with steam and found that soluble phenolic content and antioxidant activity were enhanced. Thermal processes at a the lower temperature (70℃) can maintain food quality and inhibit the activity of most enzymes [16]. In addition, ultrasound, an auxiliary processing method, could improve the texture, sensory quality, the content of organic taste compounds, and inactivate microorganisms [17-19]. Ganjdoost [20] et al. processed Agaricus Bisporus in an ultrasonic water bath and found that the quality was improved and the extended shelf life was extended. Mushroom slices were treated with ultrasound (28 kHz, 600 W), which reduced the drying time by 21.43% and obtained a superior texture [21]. Ultrasound combined thermal processes at the temperature of 70℃ was a potential method and rarely used in the processing of T. Fuciformis.” has been added in the introduction section.

  1. Terefe, N. S., Buckow, R., & Versteeg, C. Quality-related enzymes in fruit and vegetable products: effects of novel food processing technologies, Part 1: High-pressure processing. Critical Reviews in Food Science and Nutrition, 2014, 54(1), 24-63. doi:10.1080/10408398.2011.566946
  2. Wang, H. S., Ma, Y. L., Liu, L., Liu, Y., & Niu, X. D. Incorporation of clove essential oil nanoemulsion in chitosan coating to control Burkholderia gladioli and improve postharvest quality of fresh Tremella Fuciformis. Lwt-Food Science and Technology, 2022, 170. 114059. doi:10.1016/j.lwt.2022.114059.
  3. Deeth, H. C. Heat-induced inactivation of enzymes in milk and dairy products. A review. International Dairy Journal, 2021, 121, 105104. doi:10.1016/j.idairyj.2021.105104

Q2.The effect of temperature on enzyme activity is also well known and perhaps you can discuss this in relation to thermal and non-thermal treatments and how these affect colour, texture and sensory properties.

Response: Thanks for the reviewer’s comment. Those comments are valuable and very helpful. There is no doubt that enzyme activity is very important on the quality of the food during storage. it affects the color, texture, and sensory, such as polyphenol oxidase (PPO), chlorophyllase, peroxidase (POD), lipoxygenase (LOX), and lipase may be responsible for color and flavor changes, pectinases, cellulase, and hemicellulase may cause textural degradation, thiaminase and oxidative enzymes such as ascorbic acid oxidase, PPO, and POD may cause loss of nutritional value. BWB and HTS are the most common methods of thermal processes, which inhibit the activity of enzyme. Thermal processes at the lower temperature (70℃) inhibit the activity of most enzymes. Therefore, T. Fuciformis treated with boiling water blanching (BWB), high temperature steam (HTS) and the ultrasonic-low temperature blanching (ULTB, 70℃) were in the state of enzyme inactivation in our study. In this experiment, the three processing methods were the main factors that affect the color, texture, polysaccharide content and sensory characteristics of T. Fuciformis, while the effect of enzymes is negligible. So we did not investigate the effect of enzymes on the quality of T. Fuciformis in this paper. We appreciate for reviews’ warm work earnestly, and hope that the reason will meet with approval.

Q3.One question I do have about the methodology is why only 10 people were used for the sensory analysis ? The sample size seems to be very small. Could you elaborate on the training that these individuals were given please ?

Response: Thanks for the reviewer’s comment. According to the reviewer’s comments, we explain the principle of sensory evaluation. The number of sensory evaluation panelists meets the requirements of national standards "GB/T 16291.1-2012" and "GB/T 15549-1995", and the sensory evaluation members have been professionally trained. Before the sensory evaluation, they had learned the T. Fuciformis standard "NY/T 834-2004" and scoring criteria (Supplementary Table S1), including the color, texture, aroma, appearance and flavor.

Q4.Can you please look at the graphs and tables, and pay attention to the statistical analysis conducted- re-run the analysis. There appears to be some results where the values of the readings appear to be significantly different, but the analysis does not show this up. Please pay careful attention to this and when describing the results please concentrate on only those which are significantly different.

Response: We apologize for confusing you with the incorrect charts in the manuscript. We had redrawn some graphs (Figure 1, Figure 2 and Figure 3) and tables(Table 1 and Table 2,). We appreciate for reviews’ warm work earnestly, and hope that the rewise will meet with approval.
